

# Trans-oral robotic surgery versus coblation tongue base reduction for obstructive sleep apnea syndrome

Wei-Che Lan[1], Wen-Dien Chang[2], Ming-Hsui Tsai[1,3,4] and Yung-An Tsou[1,3,4]

[1] Department of Otolaryngology Head and Neck Surgery, China Medical University Hospital, Taichung, Taiwan
[2] Department of Sport Performance, National Taiwan University of Sport, Taichung, Taiwan
[3] School of Medicine, China Medical University, Taichung, Taiwan
[4] Department of Audiology and Speech-Language Pathology, Asia University, Taichung, Taiwan

## ABSTRACT

**Objectives**. To compare the efficacy of trans-oral robotic surgery (TORS) with that of coblation assisted tongue base reduction surgery in patients with obstructive sleep apnea syndrome (OSAS).

**Subjects and Methods**. The medical charts were retrospectively reviewed for all OSAS patients admitted to one institution for surgical intervention between 2012 and 2017. We analyzed 33 cases; 16 patients received TORS and 17 received coblation surgery for tongue base reduction. Both groups received concomitant uvulopalatoplasty. Surgical outcomes were evaluated by comparing the initial polysomnography (PSG) parameters with the follow-up PSG data (at least 3 months after the surgery). Epworth sleepiness scale (ESS) and complications were also compared between the 2 groups.

**Results**. The success rate ($\geq$50% reduction of pre-operative AHI and post-operative AHI <20) in the TORS group and coblation group were 50% and 58%, respectively, and there was no significant difference ($p = .611$). The AHI (mean $\pm$ SD) reduction in the TORS and coblation groups were 24.9 $\pm$ 26.5 events/h and 19.4 $\pm$ 24.8 events/h, respectively; the between-group difference was not significant ($p = .631$). ESS improvement did not differ significantly between the TORS and coblation groups (3.8 $\pm$ 6.6 and 3.1 $\pm$ 9.2, respectively, $p = .873$). The rates of minor complication were higher in the TORS group (50%) than that of the coblation group (35.3%) without statistical significance ($p = .393$).

**Conclusion**. TORS achieved comparable surgical outcomes compared to coblation assisted tongue base reduction surgery in OSAS patients. Multilevel surgery using either TORS or coblation tongue base reduction combined with uvulopalatoplasty is an effective approach for the management of OSAS.

Corresponding author
Yung-An Tsou,
D6638@mail.cmuh.org.tw,
d22052121@gmail.com

# INTRODUCTION

Obstructive sleep apnea syndrome (OSAS) is a common disorder which affects 3–7% of adult men and 2–5% of adult women (*Punjabi, 2008*). OSAS results from upper airway collapse during sleep. Clinical symptoms include fragmented sleep and excessive daytime

sleepiness (*Semelka, Wilson & Floyd, 2016*). Continuous positive airway pressure (CPAP) is thought to be the gold standard treatment for OSAS, (*Lucia Spicuzza, 2015*) but some patients cannot tolerate it and may seek surgical treatment instead (*Kotecha & Hall, 2014*). Different levels and degrees of obstruction in OSAS patients lead to variable response to surgical intervention (*Koutsourelakis et al., 2012*). In one study, *Vroegop et al. (2014)* analyzed the upper airway collapse patterns in patients with sleep disordered breathing by using drug-induced sleep endoscopy (DISE) and multilevel collapse was disclosed in 68.2% of all patients. As the intricacies of airway collapse are better understood, due to improvements in diagnostic and evaluative methods, multilevel surgery is becoming a more common method of successfully treating OSAS (*Thaler et al., 2016*; *Lin et al., 2017*). Among these patients with multilevel collapse, the most frequently seen pattern was the concomitant collapse of palatal and tongue base (25.5%) (*Vroegop et al., 2014*). Uvulopalatopharyngoplasty (UPPP) is the most commonly reported surgery to address oropharyngeal obstruction. For dealing with tongue base obstruction, trans-oral robotic surgery (TORS) and coblation assisted tongue base reduction surgery were two of the most published tongue base tissue reduction procedures.

Several *preoperative assessment* strategies have been used. The Friedman tonsil grading scale classifies the tonsil size into five grades (grade 0–IV) according to the location the tonsil relative to the surrounding structures (*Friedman et al., 1999*). The Friedman tongue position (FTP) grading system is evaluated similarly to the modified Mallampati classification, but the tongue is evaluated in a neutral position without protrusion. The Friedman staging system incorporates FTP, Friedman tonsil grading scale and BMI to classify OSAS patients into four stages: stage I includes patients with tonsils graded III–IV, FTP graded I–II and BMI $< 40$ kg/M$^2$; stage III includes patients with tonsils graded 0-II, FTP graded III–IV and BMI $< 40$ kg/M$^2$; stage IV includes patients with BMI $> 40$ kg/M$^2$ or significant craniofacial or other anatomic abnormalities; stage II includes patients beyond stage I, III, IV (*Friedman, Salapatas & Bonzelaar, 2017*). Fiberoptic nasopharyngoscopy with Muller's maneuver, which mimics the pathophysiological status of OSAS during wakefulness by asking the patient to block bilateral nostrils and inhale with mouth closed, can identify the level and degree of upper airway collapse (*Terris, Hanasono & Liu, 2000*). Drug-induced sleep endoscopy (DISE), which is recognized as a breakthrough in evaluation of OSAS patients, can provide direct identification of airway collapse during intravenous anesthesia. The VOTE classification is utilized for the findings of DISE (*Kezirian, Hohenhorst & de Vries, 2011*).

For most patients with oropharyngeal obstruction, uvulopalatopharyngoplasty (UPPP) is one of the most common and effective surgical procedures (*Khan et al., 2009*). However, oropharyngeal obstruction combined with tongue base obstruction is recognized as an important reason for failure after pharyngoplasty procedures (*Choi et al., 2016*). For tongue base obstruction, multiple procedures have been proposed and could be simply categorized into tongue base volume reduction and tongue suspension. Among these procedures, trans-oral robotic surgery (TORS) and coblation assisted tongue base reduction surgery proved to be the most published therapeutic methods in the field of tongue base reduction (*Cammaroto et al., 2017*). TORS can provide a 3D visual field and the operator can easily

**Table 1  Inclusion criteria.**

≥ 18 years old

Symptoms of obstructive sleep apnea syndrome (snoring, disrupted sleep, daytime sleepiness)

Preoperative AHI >10

Friedman tongue position grade 3 or 4

Partial or complete retropalatal and retroglossal collapse in Muller's maneuver and DISE

Cannot tolerate CPAP

**Notes.**

AHI, Apnea-Hypopnea Index; DISE, Drug-Induced Sleep Endoscopy; CPAP, Continuous positive airway pressure.

access the tongue base area and perform surgery using delicately controlled robotic instruments. Nevertheless, the high cost of TORS makes operators and patients hesitant to use it (*Cammaroto et al., 2017*). Endoscopic coblation assisted tongue base reduction surgery has been reported to be a useful procedure for tongue base obstruction and it has a lower cost compared to TORS (*Friedman et al., 2012*; *Li, Lee & Kezirian, 2016a*; *Li, Lee & Kezirian, 2016b*). However, there is a lack of fair comparison studies regarding the treatment efficacy and safety between TORS and coblation assisted tongue base reduction. Therefore, this study was conducted to compare the subjective and objective outcomes of TORS with endoscope-guided coblation tongue base reduction.

## MATERIAL AND METHODS

Medical charts were retrospectively reviewed for OSAS patients admitted for TORS or coblation tongue base reduction surgery to a single tertiary hospital between 2012 and 2017. 33 patients with age ranging from 18 to 62 years met the inclusion criteria (Table 1). Patients who were excluded were those without available postoperative polysomnography (PSG) data. PSG was performed at 3-12 months after the surgery. Patients who had previous upper airway surgery for OSAS were also excluded. This study was approved by the Institutional Review Board of the China Medical University Hospital (project *approval number* CMUH103-REC1-078).

Detailed profiles were constructed for each patient and included the following variables: age, sex, body mass index (BMI), tonsil grade, Friedman tongue position, Friedman stage, pre-operative and post-operative Epworth sleepiness scale (ESS). Post-operative ESS was recorded at the date for post-operative PSG. Nasopharyngoscopy with Muller's maneuver and drug-induced sleep endoscopy (DISE) were performed in all patients to evaluate the site of obstruction and the pattern of the airway collapse. The grades of airway collapse in Muller's maneuver were divided into four grades according to the percentage change in cross-sectional area: grade I $\leqq$25% collapse, grade II >25% and $\leqq$50% collapse, grade III >50% and $\leqq$75% collapse, grade IV >75% collapse. VOTE classification was utilized for reporting DISE findings and the grade of collapse were classified as 0 (<50% obstruction); 1 (50–75% obstruction) and 2 (>75% obstruction). Patients undergoing surgery must have at least partial tongue base collapse confirmed by Muller's maneuver and DISE. Details were also recorded from pre-operative and post-operative PSG data, and

**Table 2  Demographics, Baseline data of the 2 groups.[a]**

| | TORS group (n = 16) | Coblation group (n = 17) | p value |
|---|---|---|---|
| age, years | 39.4 ± 12.3 | 38.7 ± 11.5 | .861 |
| Male, n (%) | 15 (93.8) | 13 (76.5) | .335 |
| BMI, kg/m$^2$ | 28.2 ± 3.8 | 27.4 ± 5.6 | .645 |
| Tonsil grade | 2.0 ± 1.3 | 1.9 ± 0.8 | .748 |
| FTP | 3.4 ± 0.6 | 3.4 ± 0.6 | .918 |
| Friedman stage | 2.5 ± 0.6 | 2.7 ± 0.6 | .340 |
| Grade of collapse in Muller maneuver | | | |
| Retropalatal area | 3.2 ± 0.9 | 3.6 ± 0.6 | .147 |
| Retroglossal area | 2.6 ± 0.9 | 2.3 ± 0.8 | .260 |
| Grade of collapse in DISE | | | |
| Velum | 1.9 ± 0.3 | 1.8 ± 0.4 | .428 |
| Oropharynx | 1.5 ± 0.5 | 1.6 ± 0.5 | .624 |
| Tongue base | 1.4 ± 0.5 | 1.5 ± 0.5 | .611 |
| Epiglottis | 0.5 ± 0.6 | 0.4 ± 0.5 | .460 |
| ESS | 11.1 ± 4.7 | 10.9 ± 5.2 | .917 |
| AHI, events/hour | 50.5 ± 19.6 | 44.9 ± 28.8 | .517 |
| AI, events/hour | 33.3 ± 19.2 | 31.3 ± 26.8 | .816 |
| Min-SpO2, % | 73.8 ± 6.8 | 74.0 ± 10.0 | .951 |
| CT90, % | 15.1 ± 14.5 | 13.1 ± 15.3 | .705 |

**Notes.**

BMI, body mass index (weight in kilograms devided by height in meters squared); FTP, Friedman tongue position; DISE, Drug-Induced Sleep Endoscopy; ESS, Epworth Sleepiness scale; AHI, Apnea-Hypopnea index; AI, Apnea index; Min-SpO2, minimum oxygen saturation; CT90, cumulative time percentage with SpO2 <90%.

[a]All values are presented as mean ± standard deviation.

included AHI, apnea index (AI), lowest oxygen saturation (min-SpO2), and cumulative time percentage with SpO2 <90% (CT90) (Table 2). The success of the surgery was defined as achievement of ≥50% reduction of pre-operative AHI and a post-operative AHI <20. Perioperative parameters, including the length of stay in hospital, the numeric rating scale (NRS) assessment on the first postoperative day for pain intensity and complications, were recorded.

In this study, 16 patients received TORS and 17 patients received coblation surgery for tongue base reduction. The study flow diagram is shown in Fig. 1. All patients received conventional uvulopalatopharyngoplasty combined with tongue base reduction for multilevel obstruction in these patients.

All of the surgeries were performed by a single surgeon.

The surgical procedure of trans-oral robotic surgery for tongue base volume reduction was performed similar to the previous published literature (*Friedman et al., 2012*; *Friedman et al., 2012*). General anesthesia was introduced via nasotracheal intubation. The anesthesia machine was positioned at the left side foot of the bed. The surgical cart of the da Vinci surgical system (Intuitive Surgical, Sunnyvale, CA, USA) approached the patient from the right-hand side with an angle of 45 degrees to the bed. The scrub nurse stood next to the patient's left hand and the first assistant sat at the head of the bed. The operative
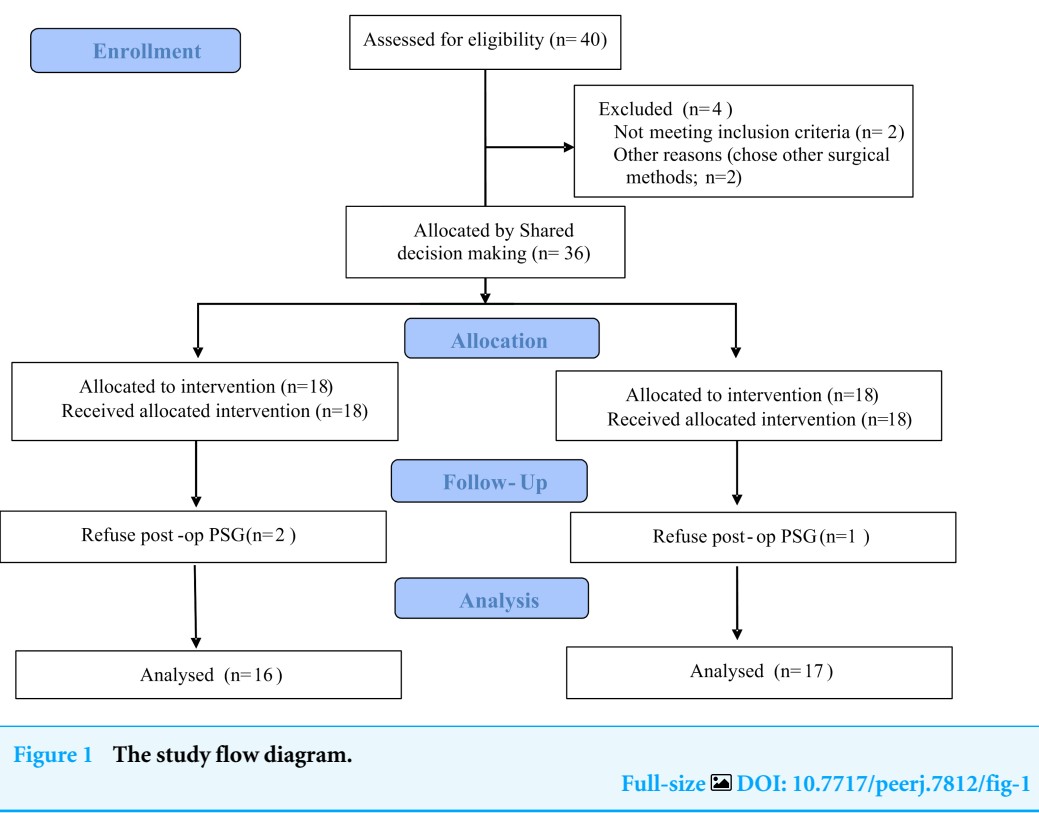

**Figure 1** **The study flow diagram.**

surgeon was at the operative console and used open-surgery hand movements which were precisely replicated in the operative field by the robotic instruments. The laryngeal advanced retractor system (Fentex, Tuttlingen, Germany) was used to expose the tongue base area. The size of the tongue blade was chosen accordingly to well expose the tongue base. Under 30 degree 3D camera endoscope, tongue base tissue was grasped by robotic forceps and cauterized with *spatula monopolar* electrode. The midline posterior glossectomy began from the foramen cecum and advanced posteriorly to vallecula without injury to epiglottis mucosa, laterally to 1 cm from the midline and 1.5 cm inferior to the tongue base surface.

Endoscopic coblation assisted tongue base reduction surgery was performed similar to previous reports (*Wee et al., 2015*; *Li, Lee & Kezirian, 2016a*; *Li, Lee & Kezirian, 2016b*). Under general anesthesia with nasotracheal intubation, the Molt mouth gag (**Sklar,** West Chester, PA, USA) was applied to the left side of labial commissure. We placed a silk suture through the anterior tongue and the silk was held by a Kelly forceps. The first assistant could easily retract the tongue forward by holding the Kelly forceps. A 70 degree rigid endoscope (Karl Storz, Tuttlingen, Germany) was applied to expose the tongue base area and kept in position by an endoscope holder (Karl Storz, Tuttlingen, Germany). With the aid of the endoscope holder, the surgeon could perform the procedure bimanually and thus decrease the operation time and the morbidity related to blood loss. The Coblator II ENT Surgery System and PROCISE MAX coblation wand (Arthrocare ENT, Sunnyvale, CA, USA) were used for the midline posterior glossectomy. The targeted resection area was the same as the TORS mentioned above.

We used the Statistical Packages for the Social Sciences version 24.0 (IBM Corp.; Armonk, NY, USA) for statistical analysis of the data. The descriptive statistic was used to present the outcome values. The Mann–Whitney and signed Wilcoxon test were used for comparing numerical variables between and within groups, respectively. Fisher's exact test was used for categorical variables. A $p$ value of less than .05 was considered to be statistically significant.

## RESULT

Among the 33 patients in this analysis, 16 were of the TORS group (age of $39.4 \pm 12.3$ years; mean $\pm$ SD) and 17 patients were of the coblation group (age of $38.7 \pm 11.5$ years, mean $\pm$ SD). Male comprised 93.8% in the TORS group and 76.5% in the coblation group. The mean body mass index (BMI) at the time of admission was $28.2 \pm 3.8$ kg/m$^2$ in the TORS group and $27.4 \pm 5.6$ kg/m$^2$ in the coblation group. There were no significant differences in tonsil grading scale, Friedman tongue position and Friedman staging system between the two groups before surgery. The grades of collapse in Muller's maneuver and DISE were similar in both groups. The mean Epworth sleepiness scale (ESS) was $11.1 \pm 4.7$ in the TORS group and $10.9 \pm 5.2$ in the coblation group. All patients received polysomnography (PSG) for pre-operative evaluation. The mean baseline apnea-hyponea index (AHI) was $50.5 \pm 19.6$ events/h and mean apnea index (AI) was $33.3 \pm 19.2$ events/h in the TORS group; corresponding values were $44.9 \pm 28.8$ events/h and $31.3 \pm 26.8$events/h, respectively, in the coblation group. The mean lowest oxygen saturation (min-SpO2) was $73.8 \pm 6.8\%$ and mean cumulative time percentage with SpO2<90% (CT90) was $15.1 \pm 14.5\%$ in the TORS group; corresponding values were $74.0 \pm 10.0\%$ and $13.1 \pm 15.3\%$ in the coblation group. Demographics and baseline PSG data for both groups are summarized in Table 2. There were no significant between-group differences prior to treatment.

The comparisons within-group (Table 3) and between-group (Table 4) were analyzed, respectively.Statistically significant improvement of ESS was observed in both groups. ESS (mean $\pm$ SD) improvement did not differ significantly between the TORS and coblation groups ($3.8 \pm 6.6$ and $3.1 \pm 9.2$, respectively, $p = .873$; 95% CI $[-3.30–2.08]$, Fig. 2). The AHI (mean $\pm$ SD) reduced significantly from $50.5 \pm 19.6$ events/h to $25.5 \pm 19.5$ events/h in the TORS group ($p = .002$). In the coblation group, the mean AHI reduced significantly from $44.8 \pm 28.8$ events/h to $25.5 \pm 23.3$ events/h ($p = .005$). The AHI reduction in the TORS and coblation groups were significantly reduced, and the between-group difference was not significant ($p = .631$; 95% CI $[-12.67–23.73]$, Fig. 3 and 4).The mean AI reduced significantly in both TORS and coblation group ($p = .014$ and $p = .004$, respectively), but the mean AI reduction did not differ significantly between the groups as well ($p = .657$; 95% CI $[-18.53–14.60]$, Fig. 4). The mean min-SpO2 improved from $73.8 \pm 6.8\%$ to $83.8 \pm 5.6\%$ ($p = .001$) in the TORS group and from $74.0 \pm 10.0\%$ to $80.7 \pm 12.6\%$ in the coblation group ($p = .045$). The mean improvement of min-SpO2 was $10.0 \pm 7.7\%$ in the TORS group and $6.7 \pm 12.6\%$ in the coblation group. There were no statistically significant differences in the improvement of min-SpO2 between the two groups ($p = .363$; 95% CI $[-4.11–10.82]$, Fig. 5). The TORS group patients had a greater reduction in CT90

**Table 3   Within-group comparison of the treatment outcomes.[a]**

|  | TORS group (n = 16) | | | Coblation group (n = 17) | | |
|---|---|---|---|---|---|---|
|  | Preoperative | Postoperative | p value | Preoperative | Postoperative | p value |
| AHI | 50.5 ± 19.6 | 25.5 ± 19.5 | .002[*] | 44.8 ± 28.8 | 25.5 ± 23.3 | .005[*] |
| AI | 33.3 ± 19.2 | 16.5 ± 17.5 | .014[*] | 31.4 ± 26.9 | 12.7 ± 22.0 | .004[*] |
| ESS | 11.1 ± 4.7 | 7.6 ± 3.6 | <.001[*] | 10.9 ± 5.2 | 8.1 ± 5.5 | .017[*] |
| Min-SpO2 | 73.8 ± 6.8 | 83.8 ± 5.6 | <.001[*] | 74.0 ± 10.0 | 80.7 ± 12.6 | .045[*] |
| CT90 | 15.1 ± 14.5 | 5.7 ± 7.6 | .005[*] | 13.2 ± 15.2 | 8.2 ± 18.3 | .183 |

Notes.

AHI, Apnea-Hypopnea index; AI, Apnea index; ESS, Epworth Sleepiness scale; Min-SpO2, minimum oxygen saturation; CT90, cumulative time percentage with SpO2 <90%.

[a] All values are presented as mean ± standard deviation.

[*] $p < .05$ is considered statistically significant.

**Table 4   Between-groups comparison of the treatment outcomes.**

|  | TORS group (n = 16) | Coblation group (n = 17) | p value |
|---|---|---|---|
| AHI reduction (events/h) | 24.9 ± 26.5 | 19.4 ± 24.8 | .631 |
| AI reduction(events/h) | 16.7 ± 23.9 | 18.7 ± 22.7 | .657 |
| ESS reduction | 3.8 ± 6.6 | 3.1 ± 9.2 | .873 |
| Min-SpO2 improvement | 10.0 ± 7.7 | 6.7 ± 12.6 | .363 |
| CT90 reduction | 9.3 ± 11.4 | 4.9 ± 14.7 | .510 |
| Success rate, n (%) | 8(50.0) | 10(58.8) | .611 |
| Day 1 pain score(NRS) | 2.8 ± 0.9 | 2.5 ± 0.7 | .533 |
| Hospital stay (days) | 5.5 ± 1.2 | 4.4 ± 0.7 | .004 |
| Major complication, n (%) | 0(0) | 0(0) |  |
| Minor complication, n (%) | 8(50.0) | 6(35.3) | .393 |

Notes.

SD, standard deviation; AHI, Apnea-Hypopnea index; AI, Apnea index; ESS, Epworth Sleepiness scale; Min-SpO2, minimum oxygen saturation; CT90, cumulative time percentage with SpO2 <90%; NRS, numerical rating scale.

percentage, but the difference was not significant. The success rate in the TORS group and coblation group were 50% and 58%, respectively, and there was no statistically significant difference ($p = .611$).

The mean numeric rating scales (NRS) for pain evaluation on the first postoperative day were similar in both groups (TORS group = 2.8 ± 0.9; coblation group = 2.5 ± 0.7; $p = .533$). In the TORS group, the length of stay in hospital was longer compared with the coblation group ($p = .004$). There were no major complications (e.g., intra-operative or post-operative bleeding, airway compromise, prolonged intubation, pneumonia and pharyngeal laceration, tongue limitation) in either group. No tracheotomies were performed for airway management perioperatively. The rates of minor complication, including transient dysphagia, pharyngeal edema and dysgeusia, in the TORS and coblation groups were 8/16 (50%) and 6/17(35.3%), respectively.

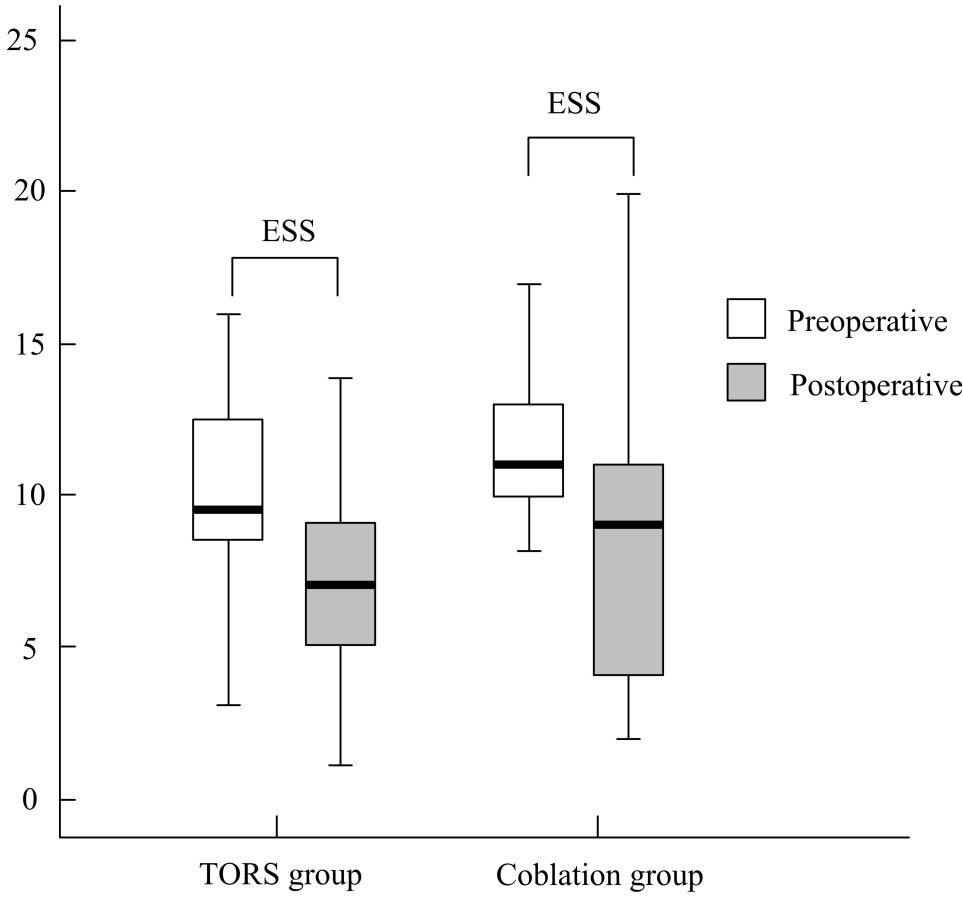

**Figure 2** The treatment outcome of ESS between two groups. .

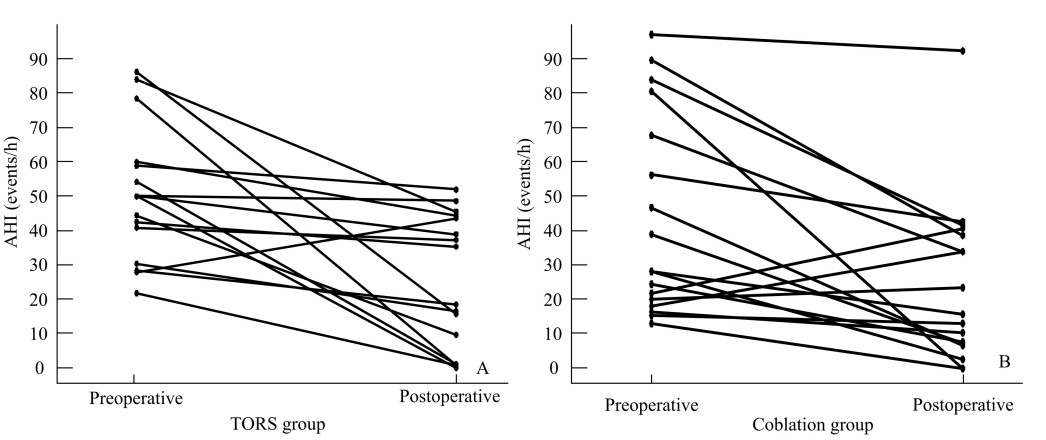

**Figure 3** Individual AHI decrease in TORS (A) and Coblation (B) groups.

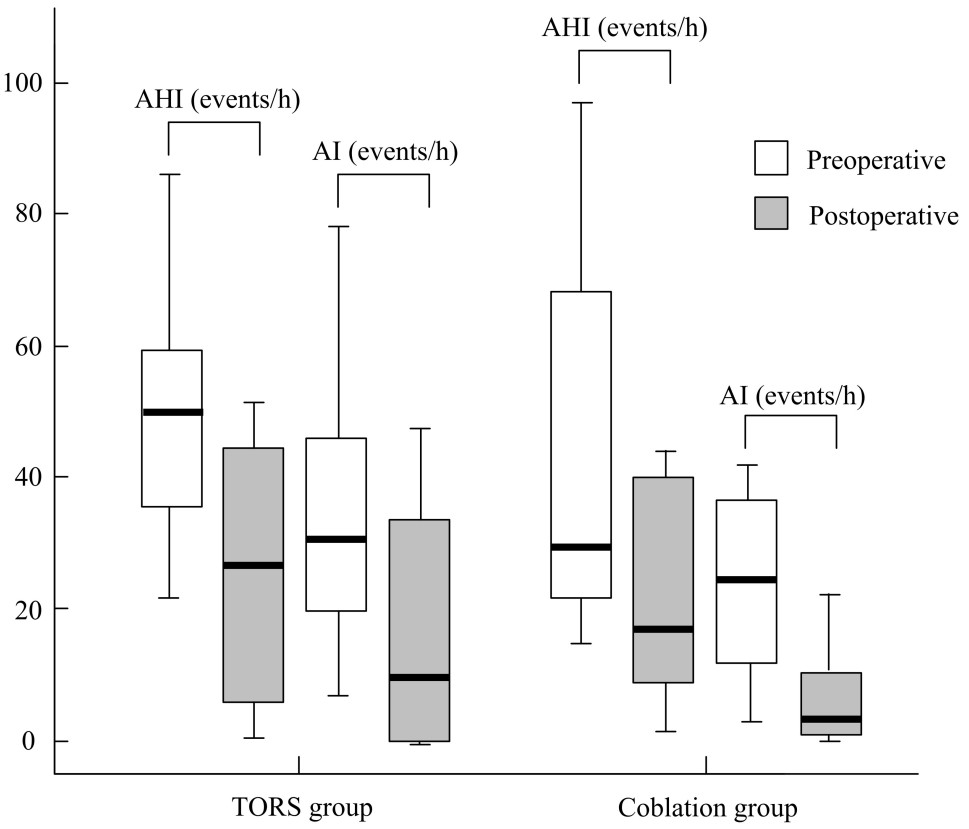

**Figure 4 The treatment outcome of AHI and AI between two groups.**

## DISCUSSION

Our results presented that the surgical outcomes of trans-oral robotic surgery (TORS) were comparable to coblation assisted tongue base reduction surgery in obstructive sleep apnea syndrome (OSAS) patients. The PSG outcomes and success rate were similar for the TORS and coblation groups.

Multilevel surgery is thought to be more effective than UPPP for management of patients suffering from OSAS. Because of the better understanding of the complexity of the upper airway collapse during sleep in OSAS patients, surgeons can determine correct surgical management according to the site of obstruction and the pattern of the airway collapse (*Toh et al., 2014*; *Li, Lee & Kezirian, 2016a*; *Li, Lee & Kezirian, 2016b*; *Thaler et al., 2016*).

In a retrospective study, 25 moderate-to-severe OSAS patients with retropalatal and tongue base obstruction received coblation endoscopic lingual lightening and modified uvulopalatopharyngoplasty (relocation pharyngoplasty). AHI (mean $\pm$ SD) decreased significantly from $45.7 \pm 21.7$ to $12.8 \pm 8.2$ events/hour ($p < .001$) postoperatively and the overall surgical success rate was 80% (*Li, Lee & Kezirian, 2016a*; *Li, Lee & Kezirian, 2016b*). The Coblation lingual tonsil removal technique proved to be an effective procedure in a cohort of Korean OSAS patients with retroglossal obstruction. The mean AHI decreased significantly from $37.7 \pm 18.6$ to $18.7 \pm 14.8$ events/hour ($p < .001$) and the success rate

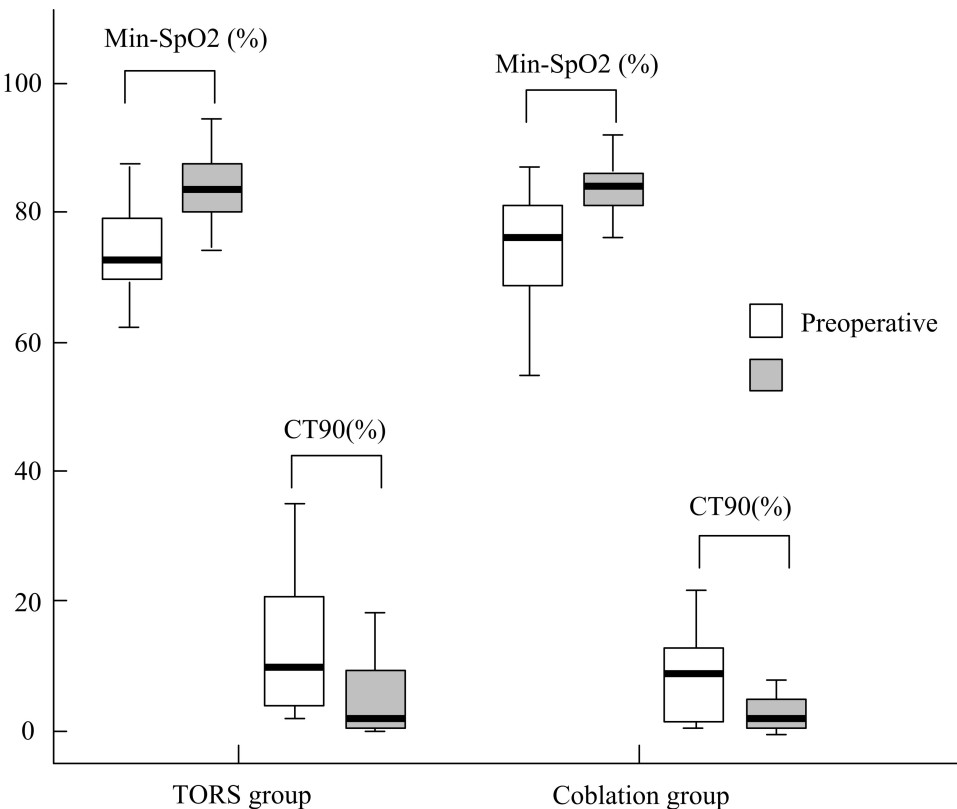

**Figure 5** The treatment outcome of min-SpO2 and CT 90 between two groups.

was 55.6% (*Wee et al., 2015*). Another study compared combined coblation endoscopic lingual lightening and relocation pharyngoplasty to relocation pharyngoplasty alone in OSAS patients (AHI >20, Friedman stage III), and reported that combined surgery had better improvement in AHI (−65.5 vs −53.2; $p = .047$) and higher surgical success rate than relocation pharyngoplasty alone (73% vs 50%; $p = .04$) (*Li, Lee & Kezirian, 2016a*; *Li, Lee & Kezirian, 2016b*).

*O' Malley Jr et al. (2006)* developed a minimally invasive surgical procedure for management of tongue base neoplasms by using robotic surgical instruments. Trans-oral robotic surgery (TORS) can offer clear 3D visualization and gain adequate access to tongue base, larynx and hypopharynx and provide meticulous tissue resection. A preliminary study in 2010 conducted by Vicini et al. reported that TORS for tongue base resection in OSAS patients is practical and well tolerated. Ten patients were included and the AHI (mean ± SD) decreased from 38.3 ± 23.5 to 20.6 ± 17.3 events/hour (*Vicini et al., 2010*). Further study for demonstration of the feasibility of TORS performed in forty four patients with OSAS reported significant improvement of mean AHI (24.6 ± 22.2 events/hour) and mean ESS (5.9 ± 4.4) (*Vicini et al., 2012*). The latest systematic review and meta-analysis by *Meccariello et al. (2017)* concluded that TORS seems to be a promising and safe technology for the management of OSAS and the mean failure rate was 34.4% (29.5–46.2%).

A study by Friedman et al. in 2012 was thought to be the first comparative study for comparison of coblation and TORS in OSAS treatment (*Friedman et al., 2012*). It compared the effectiveness of TORS with that of coblation assisted submucosal minimally invasive lingual excision (SMILE). All the patients in the study received concomitant z-palatoplasty. The AHI (mean ± SD) reduction in the TORS and SMILE groups were 60.5% ± 24.9% and 32.0% ± 43.3% ($p = .012$), respectively. The success rate in the TORS and SMILE groups were 66.7% and 45.5%, respectively; the between-group difference was not significant ($p = .135$). However, the techniques used by each group were different and not completely comparable.

To the best of our knowledge, there is few finely matched studies regarding the treatment efficacy and safety between TORS and coblation adopting similar technique in tongue base resection. Our retrospective comparison of TORS with coblation in the treatment of OSAS patients with multilevel obstruction found that both groups had similar surgical results.

The demographics and preoperative polysomnographic data did not differ significantly between the two groups (Table 2) at baseline. According to the within-group outcomes showed in Table 3, statistically significant improvement of apnea-hyponea index (AHI), apnea index (AI), Epworth Sleepiness scale (ESS) and minimum oxygen saturation (min-SpO2) were noted in both the TORS and coblation groups. It confirmed that either TORS tongue base resection or coblation assisted tongue base resection combined with concomitant uvuolopalatoplsty can offer reliable surgical results. The cumulative time percentage with SpO2 <90% (CT90) were decreased in both groups but only significantly reduced in the TORS group which could be related to small sample size or poor correlation of CT90 to AHI (*Chung et al., 2012*).

As detailed in Table 4, the mean reduction of AHI, AI, ESS, CT90 and mean improvement of min-SpO2 were similar for the TORS and coblation groups. The rate of surgical success in the TORS group were comparable to the coblation group (50% vs 58%, $p = .611$). *Hwang et al. (2018)* compared the tongue base coblation resection to TORS in OSAS patients and both groups were in combination with lateral pharyngoplasty. They reported that the surgical success rates did not differ significantly between the two groups (56.3% in TORS vs 62.1% in coblation, $p = .711$). Our success rates are lower than those in that study. However, the preoperative BMI of patients in that study was lower than in our study group (TORS group = 28.2 ±3.8 kg/m$^2$; coblation group = 27.4 ± 5.6 kg/m$^2$, $p = .645$). Moreover, preoperative mean ESS were lower (TORS group = 11.1 ± 4.7; coblation group = 10.9 ± 5.2, $p = .917$) and mean min-SpO2 were higher (TORS group = 73.8 ± 6.8%; coblation group = 74.0 ± 10.0%, $p = .951$) in their study than those in our study group, which might suggest the *severity* of *OSAS is greater* in our patients. In our study, the mean pain scores (numeric rating scales) on the first postoperative day were comparable in both groups (TORS group = 2.8 ± 0.9; coblation group = 2.5 ± 0.7; $p = .533$). In the TORS group, the length of stay in hospital was longer compared with the coblation group ($p = .004$). There was no major complication in either group. The rates of minor complication were higher in the TORS group (50%) than that of the coblation group (35.3%) without statistical significance. According to a review article, slightly better outcomes were observed in TORS compared to coblation, but the higher rate of minor

complications and the significant costs of TORS are two aspects which surgeons will need to consider (*Cammaroto et al., 2017*).

This study has some limitations. First, the retrospective analysis used in this study is a possible source for selection bias by patients' preferences even after shared decision makings and routine surgical treatment strategy explanation. Second, it is difficult to make comparisons among studies because of different surgical techniques utilized by TORS (e.g., midline posterior glossectomy (*Lin et al., 2015*), lingual tonsillectomy (*Muderris et al., 2015*)) and coblation (e.g., midline posterior glossectomy (*Li, Lee & Kezirian, 2016a*; *Li, Lee & Kezirian, 2016b*), SMILE (*Friedman et al., 2008*), *channelling* of the *tongue* (*Zhang & Liu, 2014*), Interstitial injections with needle coblation (*Hou, Hu & Jiang, 2012*)). Third, non-parametric approach is that we are unable to adjust for potential confounders. In the future, prospective, randomized, controlled trials that incorporate similar surgical technique will be needed to evaluate the efficacy of TORS compared with coblation tongue base reduction. Moreover, studies providing long-term results in the treatment of OSAS are also warranted.

## CONCLUSION

TORS resulted in *comparable* objective and subjective outcomes compared to coblation assisted tongue base reduction surgery in OSAS patients. Multilevel surgery with either TORS or coblation tongue base reduction combined with uvulopalatoplasty is effective in reducing disease severity in moderate-to-severe OSAS cases.

## ACKNOWLEDGEMENTS

The authors would like to thank all colleagues of department of Otolaryngology Head and Neck Surgery in China Medical University Hospital who provided insight and expertise that greatly assisted the research.

### Funding

This work was supported by China Medical University (No. CMU-102-bc3 and DMR-107-040). The funders had no role in study design, data collection and analysis, decision to publish, or preparation of the manuscript.

### Grant Disclosures

The following grant information was disclosed by the authors:
China Medical University: CMU-102-bc3, DMR-107-040.

### Competing Interests

The authors declare there are no competing interests.
## Author Contributions

- Wei-Che Lan performed the experiments, analyzed the data, contributed reagents/materials/analysis tools, prepared figures and/or tables.
- Wen-Dien Chang analyzed the data, contributed reagents/materials/analysis tools, prepared figures and/or tables, authored or reviewed drafts of the paper.
- Ming-Hsui Tsai authored or reviewed drafts of the paper.
- Yung-An Tsou conceived and designed the experiments, performed the experiments, analyzed the data, contributed reagents/materials/analysis tools, prepared figures and/or tables, authored or reviewed drafts of the paper, approved the final draft.

## Human Ethics

The following information was supplied relating to ethical approvals (i.e., approving body and any reference numbers):

This study was approved by the Institutional Review Board of the China Medical University Hospital (project approval number CMUH103-REC1-078).

## Data Availability

The raw data of this retrospective study are available in File S1.

## Supplemental Information

Supplemental information for this article can be found online at http://dx.doi.org/10.7717/peerj.7812#supplemental-information.

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
