# Peer review of "Trans-oral robotic surgery versus coblation tongue base reduction for obstructive sleep apnea syndrome"

_PeerJ, doi:10.7717/peerj.7812_

## Round 0.1 · original submission · Major Revisions

Please respond the critiques from the reviewers

·

Basic reporting

First, I’ll make a couple of general comments about the presentation of results:
* Whenever you use ± notation (e.g. Lines 31, 33, 147, 150, etc.), please indicate what the values before and after the ± symbol are, at least the first time you use this notation in each section. These could be assumed to be mean ± SD, but they could also be mean ± SE, mean ± 95% CI half width, etc.
* Given the number of surgeries (33 overall, 16 and 17 by group), there is no need for a decimal place in percentages (e.g. Line 35). Each person overall counts for around 3% themselves, and within groups, this is around 6% each, so the values after the decimal place are not meaningful.

I will suggest some additional information be added to the abstract for the reader who might not read the entire manuscript. Specifically:
* The reader of the abstract will probably want to know when the follow-up measurements were taken (Lines 29–30, for both the follow-up AHI and ESS, and the period during which complicated were recorded). The first of these is given as 3 months+ (what was the minimum–maximum?) on Line 78, but the other time periods are not explicitly described in the manuscript (Lines 85 and 100).
* I would suggest starting the abstract results with a sentence about these patients (Line 31), including noting any potentially important differences between the surgery groups or the lack of such.
* The study aim is addressed by between-group comparisons (i.e. between TORS and coblation assisted tongue base reduction surgery) rather than within-group comparisons. While providing the pre and post values in the abstract on Lines 31–33 is potentially helpful, rather than or as well as showing within-group p-values here, please present these crucial between-group p-values for all outcomes in the abstract (currently AHI, success, and complications), rather than only for the second of these. Why is ESS not included here also?

The introduction could be improved with some additional references. Any fact or claim made here should be supported by the literature. In particular, I think references are needed for Lines 45 (symptoms), 47–48 (tolerance and surgery seeking), 49 (variable responses), 53 (increasing use of multilevel surgery), 58 (reason for failure), and 65 (hesitance to use TORS).

When reporting the results (Lines 165–186), I think it would be easier for the reader if you interleave the within- and between-group analyses so that each variable is covered in turn, rather than presenting within-group analyses for all variables and then doing the same for the between-group analyses. The between-group analyses are the ones that address your research aim and so I feel that these need to be emphasised more, with the within-group results there to support the efficacy of the surgeries more generally.

A reasonable amount of the Discussion (Lines 198–237, possibly Lines 239–250, and perhaps also Lines 253–255) could be removed or moved (in whole or in part) to the Introduction. None of this material discussed your study’s findings, the relationship of these findings with the literature, or its implications for practice or research. Note that it is tradition to begin the Discussion with a description of the main or overall pattern of findings from the results before comparing these to the literature, leading into these implications. Some, but not all, of the material on Lines 281–311 could be moved to the start of the Discussion to summarise your findings. Results from the literature that are not going to be compared to your study results in some way or another should, I think, be moved to the Introduction or deleted entirely.

Some minor specific comments about the writing are listed below:
Lines 39–40: Perhaps either “…is a successful approach for the management of OSAS.” or “…is successful for managing OSAS.” A similar comment would apply to Line 198.
Line 57: Do you mean “…as the most important…” or “…as an important…” here?
Line 58: Do mean “procedures” here?
Line 90: 50% would appear to be in both grade II and grade III here.
Line 108: I suggest just “literature” here.
Lines 117–118: The sentence here (“The curved frame of the retractor could fit with the shape patient’s face and the extended framework could make the movement of the robotic arm easily.”) needs rewriting.
Line 129: “held” rather than “hold” here?

A data dictionary would be a very useful complement to the data provided. This could be an additional worksheet within the workbook or as an external document. In either case, this would list all variables and give the meaning of the variable and the interpretation of each value for binary, ordinal, and categorical variables. For example, at the moment it is not clear whether sex=0 or sex=1 is male. I think the variable named “posai” should be named “postai”.

Experimental design

How many patients did not have follow-up data (Lines 77–78)? What were the reasons for this and did these patients differ from those with follow-up data? How many were also excluded for having had previous surgery (Line 79)? This would appear to tighten the research question to being about differences in the procedures amongst those who have not had previous surgery and, if so, this should be made clear back on Lines 24–25 and 71. You might find a patient flow diagram (like a CONSORT diagram despite the lack of randomisation here) to be helpful here in showing the missing data both overall and by surgery group.

Note that all statistical tests make some form of assumptions and these need to be explained in the statistical methods (Lines 139–142). This includes assumptions about the residuals for independent sample t-tests (normality and homoscedasticity) and paired sample t-tests (normality of delta values, note also that this is the usual nomenclature rather than “dependent t-test” on Lines 141 and 165), and most statisticians require expected cell counts to exceed 5 in at least 80% of cells for a Chi-squared test. For the t-tests, based on the data shown later in the tables, these assumptions will not be met here in all cases. Better than independent sample t-tests would be an ANCOVA comparing follow-up values between groups and including baseline values as a covariate. This increases power if changes are correlated with baseline values, which is often the case (the patients who are worst off might have the potential for greater improvements in absolute terms—this might be the case with CT90 in Table 3 for example). Again, if you use this approach, you will still need to discuss model checks, now including linearity of residuals with respect to baseline values. You might find natural logarithmic transformations to be helpful, or you may need to look at semi- or non-parametric alternatives.

Another advantage from using a regression-based approach such as ANCOVA is that you could adjust for potentially important variables that differ between surgery groups. While you have not at this stage identified any statistically significant differences between groups (Line 163), you might still feel that some values are numerically different enough to be worth adjusting for.

Note that means and SDs are not useful for most readers in most cases for non-normally distributed data and the values for AHI, AI, ESS, Min-SpO2, and CT90 in Table 3 are all skewed to a greater or lesser extent. You might find it more useful to present medians and IQRs in this case, or you could investigate transformations of these variables. The changes for these variables in Table 4 should also be investigated for normality.

Validity of the findings

If you wish to make statements such as “comparable surgical outcomes” (Line 37), this would require an equivalence approach to comparing the groups rather than a superiority one. You would need to define equivalence/comparable (in units for each variable) and then focus on the 95% CIs rather than the point estimates and p-values for this interpretation. Given the higher cost of TORS, I think an equivalence or a non-inferiority design would better address what I see as the primary question (can coblation assisted tongue base reduction surgery be used in place of TORS?)

Table 2 presents means and SDs for some ordinal items (stages and grades) which I would not consider valid, nor would I expect the t-tests using these to be valid due to the discreteness of the distributions. Note that the tests used should be apparent in the table itself as well as explained in the text. This last point also applies to Tables 3 and 4. Note also that using the standard approach to Chi-squared tests (80% of expected cell counts exceeding 5), Fisher’s exact test could be preferred for comparing sex between groups.

Table 3 needs to present within-group changes and 95% CIs to help make it clear what values are consistent with the data. Similarly, Table 4 needs to present between-group differences and 95% CIs for all outcomes (this will particularly help when you present and discuss the LoS result on Lines 190 and 305–306). I suggest moving the between-group comparisons for AHI reduction, AI reduction, ESS reduction, Min-SpO2 improvement, and CT90 reduction from Table 3 to Table 4, leaving the post-surgery outcomes in Table 4. Given the small sample size, if the between-group intervals are wide, this would mean that no conclusions about the outcomes from the surgeries being comparable could be made and further research is needed. On the other hand, if these intervals did not include meaningful differences, you would be able to make slightly stronger claims about the lack of differences between the surgeries, subject to selection effects for each.

An important aspect not covered here is the cost of the procedures (aside from a mention in the Introduction around Line 64–67, and Discussion around Line 310). While a full cost-effectiveness analysis might be beyond the scope of the present work, some readers will be interested in the relative cost of the two surgeries. Are you able to provide any data on this (e.g., operating theatre time alongside LoS data, ideally with standard costs used for each?)

Additional comments

While this is an interesting and important topic, there is work needed, particularly for the statistical analyses and in improving the flow of the Discussion.

Reviewer 2 ·

Basic reporting

no comment

Experimental design

no comment

Validity of the findings

no comment

Additional comments

103 Are there differences in the way of doing UPPP in those patients? Are there reposition pharyngoplasty or expansion sphincter pharyngoplasty done to these patient? Because in these multilevel surgeries, different pharyngoplasty chosen may also affect the outcome.

148 To report percentage of male number and percentage is enough, as well as in the table.

156 Please report the range of AHI values for both groups, so we could know better the patient selection of the authors in the very beginning.(Seems the authors chose the severe OSA group as their surgery target?)
Graphs for individual AHI decrease may also be helpful, because AHI values for some unsuccessful patients may increase after surgery.

165 t test may not be appropriate in this small patient volume. Non-parametric test like Mann-Whitney and Wilcoxon test are more appropriate if these values failed normal distribution test, and median values with quartiles should be reported instead of mean and standard deviation.

198 Using more effective than UPPP only rather than just successful would be more appropriate.

295 Hwan-->Hwang

326 Conclusion of abstract and main article: successful treatment is an exaggeration. 50 plus success rate is far from success. Although Sher criteria is too strict for severe OSA group, 20 plus residual AHI for about half patients does not mean success either. Better statement would be: Multilevel surgery using either TORS or coblation tongue base reduction combined with uvulopalatoplasty are effective in reducing disease severity in some severe OSA cases.

Table 1 and discussion :Are there differences of success rate or AHI reduction between patients with partial or complete retroglossal collapse in DISE? i.e. Are preoperative DISE findings of tongue affect surgery outcome?

Table 3 is a little confusing. TORS , Coblation and their related variables should be at the left vertical panel; preoperative, postoperative comparison and p values should be reported at the horizontal panel.

Table 4 To emphasize mean SD in the left panel of the table is not necessary. Just mention that in the note would be fine.

---

## Round 0.2 · Minor Revisions

The authors should fully address the comments from the reviewers.

·

Basic reporting

Thank you for addressing the ± notation in the abstract. Can you also do this in the results around Line 162 when you first use this? Some readers will of course skip the abstract and read only the manuscript body.

My requests about the data (“A data dictionary would be a very useful complement to the data provided. This could be an additional worksheet within the workbook or as an external document. In either case, this would list all variables and give the meaning of the variable and the interpretation of each value for binary, ordinal, and categorical variables. For example, at the moment it is not clear whether sex=0 or sex=1 is male. I think the variable named “posai” should be named “postai”.) do not appear to have resulted in any changes despite the positive remark in the rebuttal (“Answer: Thank you, and we supply it.”). Was the updated spreadsheet uploaded along with the revised manuscript (the last edited date on Raw_data_.xlsx is 30-April-2019 which suggests that this is the old data spreadsheet)? Can you please also check that all variables presented in the manuscript are included in the data, I couldn’t find the Friedman stage or ESS, as examples, there (my apologies if I’m overlooking them, although this is also where a data dictionary is helpful).

The statistical test used to produce each and every p-value should be evident from the table through notes in that table.



Some minor comments about the writing:

Line 62: Perhaps “THE Friedman tonsil grading…”

Line 64: Perhaps “THE Friedman tongue position…”

Line 66: Perhaps “THE Friedman staging system…”

Line 85: Perhaps “…in the field of tongue base reduction…” (deleting the second “the”).

Line 88: Suggest deleting “make” and “of” here (already use “makes” earlier in the sentence), leaving “…makes operators and patients hesitant to use it.”

Line 119: “…AND cumulative…” for the last list item.

Lines 156–157: Please explain which descriptive statistics were used here.

Line 157: Note that there are two Wilcoxon tests, the between- and within-group tests. If this is referring to the within-group test, “signed Wilcoxon test” would make this clear, even more so if “between and within groups respectively” was added at the end of the clause (just before the comma on Line 158).

Line 158: “…Fisher’s EXACT test…”

Line 164: I suggest “of the” for both instances of “in the” here. Perhaps also start the sentence (Line 163) slightly differently, perhaps “Males comprised…”.

Line 176: “Demographics AND baseline PSG data…”

Line 182: The tilde (“~”) is normally used for “approximately” rather than to indicate ranges. I suggest a hyphen (“-“) or, better, an en-dash (“–“) here instead. Also Line 187 and elsewhere.

Line 194: “…STATISTICALLY SIGNIFICANT differences…”

Lines 195–196: “…had a GREATER reduction IN…

Line 199: “mean” rather than “average”. See also Lines 224 and 277.

For the results on Lines 199–201, it would be much easier to appreciate these, especially the different in length of stay, with descriptive statistics included here. Also on Line 277–280 where values will help you indicate potential clinical significance as well as the results of a hypothesis test.

Line 206: An integer percentage would be more appropriate given the sample size. Same point on Line 268, 281, and possibly elsewhere.

Line 209: I’d avoid such a definite term as “demonstrate” here as there is room for the patients to vary between the two treatments.

Line 222: “THE coblation lingual tonsil removal technique…”

Lines 242–243: You can delete “the comparison of” as you’ve described this as a “comparative study”.

Line 251: I’m not sure what you mean by “a lack of fine and matched studies”. Do you mean “finely matched” or “high quality and matched”?

Lines 273–274: These values could be misread as min–max BMI rather than the two means (this is not stated on Line 273 as you do for other outcomes on Lines 274 and 275, for example). As there are only two means, you could list them instead here and for other outcomes.

Experimental design

As noted previously, there is substantial skew for many variables (e.g. Lines 172–173, 175, 176, many of the variables in Table 2 and 3). This makes reporting means and SDs questionable at best. It is not usual to report means and SDs, which are generally treated as parametric descriptives in that they assume a normal distribution, at least without another context being provided, and then to use non-parametric analyses. When non-parametric analyses are used, this generally indicates that means are not normally distributed, leading to medians (and the IQRs or 25th and 75th percentiles for variability).

There appear to be some rounding errors in Table 2, e.g. mean BMI for TORS is 28.15625 which rounds to 28.2 and not 28.1 as reported. Similarly, mean BMI for the coblation group should be 27.3 not 27.3 and the SD 5.6 not 5.5. I get quite different values for AI (means of 33.3 versus 34.3 and 31.3 versus 28.2). This all assumes that the uploaded data is correct of course. I have not checked for all other possible discrepancies, so I suggest that all values in Table 2 are also checked.

The statistical methods refer to Mann-Whitney U, (signed) Wilcoxon, and Fisher’s exact tests (“The Mann-Whitney and Wilcoxon test were used for comparing numerical variables, and Fisher’s test was used for categorical variables” on Lines 157–158), but the handful of p-values for non-categorical variables that I looked at in the Tables seemed to be from t-tests instead or did not match either this or the listed non-parametric test. As also noted previously, t-tests are not appropriate for discrete data, such as tonsil grades. I am unable to replicate some of the p-values using either t-tests or Mann-Whitney U tests so I am unsure what was done here.

While I can replicate some descriptive statistics and p-values in Tables 3 and 4, some of the values I get from the data are quite different. It required some trial and error for me to realise that some outcomes not indicated as percentages in Table 4 were being analysed as percentages, e.g. AHI, which was also changed to 0 in the case of worsening (this should be made clear somewhere if it is not already). Even after being able to replicate the means for AHI % change, the SD for the TORS group still differed for me (SD 38.0 not 38.9). I wonder if the uploaded data are what was used to produce the results in the manuscript.

Related to the above, the values presented on Lines 185–186 say they are events per hour, but they appear to be percentage reductions instead. Note that for the coblation group, the reduction of 45.3 “events/h” (Line 186, Table 3) is greater than the pre-surgery mean of 44.8 (Line 184, Table 2).

Validity of the findings

An important limitation that follows from using non-parametric approaches is that you are unable to adjust for potential confounders. Given the data set size, this adjustment would always have been limited in terms of the number of variables that could be accommodated. Possibly related to this, can you be more precise about the “selection bias” on Line 286 and, if possible, consider the expected direction of this bias for your results. The other major limitation of non-parametric approaches is the lack of resulting confidence intervals, which makes interpreting results, including non-statistically significant results, difficult. Even with this lack, however, I feel that more could be made of the results by adding clinical interpretation of differences. Is the 1.1 (not 1.2 as suggested in Table 4 if the uploaded data is correct) day difference in hospital stays clinically relevant or not in your view?

Additional comments

The manuscript is much improved but after noticing some confusing results (at least to me), I looked at the data and found what appears to me to be a number of issues. Some of these are minor rounding discrepancies, others are different values obtained from the uploaded data compared to the manuscript. Some variables were not clear in their meaning (e.g. the reduction in AHI events/h appears to be a percentage) and other variables seemed to be lacking in the data set and could not be examined. Some of the p-values could not be replicated using the statistical methods described, but could using alternative tests, and some could not be replicated even after investigating alternatives. As far as I can tell, none of these affects the overall findings, but I would strongly suggest the authors carefully check each and every value using the uploaded data file to ensure that everything is correct. If any of this is due to my misunderstandings, I hope that this is determined quickly and I apologise in advance.

Reviewer 2 ·

Basic reporting

no comment

Experimental design

no comment

Validity of the findings

no comment

Additional comments

One of your inclusion criteria is AHI > 20. However, there are 4 patients in your raw data did not meet this criteria(see below). Please explain, change your inclusion criteria or reselect your patients.

preAHI
1050527 14.4
1020702 16.9
1051210 17.8
1050902 19.5

[Graphs for individual AHI decrease may also be helpful, because AHI values for some unsuccessful patients may increase after surgery.]

For this question, I didn’t mean whisker bar graph. Can you provide graph like figure 1B of this paper?
DOI:10.1038/s41598-018-22710-1
It would be much easier for us to follow.

[Table 3 is a little confusing. TORS , Coblation and their related variables should be at the left vertical panel; preoperative, postoperative comparison and p values should be reported at the horizontal panel.
Answer: Yes, we had revised them.
Table 4 To emphasize mean SD in the left panel of the table is not necessary. Just mention that in the note would be fine.
Answer: Yes, we had revised them.]

For these questions, I suggest you authors should show the original version and changed version in the response letter at the same time, because it is hard for me to follow the difference in the manuscript. Actually, I would love to see you authors answer all my questions in this way.

Example(please do this in tables):
Page Line Original Revised

---

## Round 0.3 · accepted · Accept

Thank you for addressing the reviewers comments.